# Research on the Reliability of Bridge Structure Construction Process System Based on Copula Theory

Qingfu Li and Tianjing Zhang *

School of Water Conservancy Engineering, Zhengzhou University, Zhengzhou 450001, China
* Correspondence: 202012222024294@gs.zzu.edu.cn; Tel.: +86-178-0386-3022

**Abstract:** Various random factors in the bridge construction process directly affect the safety of the bridge life cycle. The existing theories on the reliability of bridge structure mainly focus on the reliability of components and the reliability of the bridge structure system in the completion and operation stages, while the research on the reliability of the structure system in the construction stage is relatively lacking. Therefore, this paper proposed using the Copula function to calculate the reliability index of the bridge structure construction process system. The basic theory of the Copula function was introduced in detail, and the formula was improved according to the actual situation of bridge construction. Finally, the sensitivity analysis of bridge system reliability was carried out. The research results showed that the method proposed in this paper based on Copula theory to calculate the reliability index of the bridge structure construction process system has strong applicability, simple calculation, and can be used in conjunction with the "interval estimation method", which is suitable for large and complex bridge structural engineering. At the same time, the conclusion that the influence of failure mode correlation on structural reliability should not be ignored in the actual engineering construction process is confirmed.

**Keywords:** bridge structure construction; Copula theory; system reliability



## 1. Introduction

In recent years, with the continuous improvement of comprehensive national strength, bridge construction in China has gradually developed towards higher pier height and larger span. At present, the common structural forms of large span complex bridges in China include but are not limited to large span steel pipe concrete arch bridges, large span suspension bridges, and steel–concrete composite continuous girder bridges. As one of the most important steps of bridge construction, the bridge construction stage directly determines the quality of the bridge after completion. Bridge construction includes many divisional works, such as foundation and substructure, superstructure, and protection works, etc. It involves a wide range, complex construction, many influencing factors, and harsh construction conditions, which are very conducive to engineering accidents. At the same time, along with the great development of bridge construction, safety problems are becoming more and more prominent. The number of overturning and collapse accidents in the construction process is gradually increasing, thus directly causing a large number of economic losses and casualties and producing adverse social impacts. Therefore, it becomes necessary to analyze the structural response during bridge construction.

A series of faults are inevitable in the process of manufacturing and use of engineering structures. It is of great significance to accurately identify the failure mode and predict the probability of failure for the extension of structural life. In this regard, many scholars at home and abroad have carried out a series of studies. Soliman M et al. [1] better predicted the fatigue life of steel bridge structures by integrating structural health monitoring into a probabilistic bilinear S-N approach. Helder Sousa et al. [2] carried out construction evaluation and long-term prediction of prestressed concrete bridges based on monitoring

data. Based on the long-term monitoring data of Lezíria Bridge, which is the new bridge on the Tagus River in Portugal, an analysis strategy for calculating the long-term behavior based on the finite element model is proposed. The research results showed that the trend caused by shrinkage and creep and the change caused by temperature can be used as the main references in the detection of the viaduct. Lei Li [3] summarized the advantages and disadvantages of the current large-scale bridge dynamic displacement monitoring methods, constructed a GPS bridge monitoring system, introduced the measurement principle, system composition, data processing method, and precautions of the system in detail, and concluded that the measurement result of the system is stable and high precision through an example analysis. Chengxin Yu et al. [4], in order to make up for the shortcomings of traditional measurement methods in monitoring the overall deformation of bridges and long-term real-time monitoring, and overcome the defect that the digital photography technology based on monocular vision cannot monitor the three-dimensional deformation of bridges, proposed a bridge-monitoring method based on image matching-time baseline parallax method, and established a new bridge-monitoring and early warning system on this basis. The superiority of this method was verified by an engineering example. Xin Wei et al. [5] developed a bridge health-monitoring data integration and early warning system using component GIS software in order to efficiently integrate, manage, and visually analyze a large amount of existing data in the bridge monitoring project, and carry out real-time dynamic early warning on its monitoring data, which greatly improved the operability and visualization level of data in the monitoring process. Ghiasi R et al. [6] used a non-probabilistic element model and interval mathematics to deal with the uncertainty in structural damage detection, and proposed a non-probabilistic agent model based on Wavelet Weighted Least Squares Support Vector Machine (WWLS-SVM) to solve the uncertainty problem in vibration-based damage detection. Through practical application, it was proven that the performance of the method was better than the direct finite element model, and less computation was required. Ashraf A. A. Beshr et al. [7] established an integrated monitoring system using global navigation satellite systems (GNSS) observation. The system is mainly used to study the deformation behavior and displacement prediction of suspension highway bridges, taking into account the effects of wind, temperature, humidity, and traffic load during operation and short-term measurement. The results showed that the predicted displacement obtained by artificial neural network (ANN) and adaptive neuro-fuzzy inference systems (ANFIS) provides a significant improvement for predicting the structural deformation of suspension highway bridges based on GNSS observations. Davide Martucci et al. [8], by applying the extreme value function theory to structural damage detection, provided a useful tool for structural health monitoring, and carried out numerical and experimental tests and verifications by using engineering application examples in different fields. This research laid a solid foundation for structural health monitoring in the field of extreme function theory (EFT) and extreme value theory (EVT) in the future. Susheng Li et al. [9], by integrating building information modeling (BIM) with health monitoring and early warning, achieved the goal of integrating bridge operation, management, and maintenance, provided visual and information-based bridge conditions for bridge management and maintenance departments, and then provided information-based data assistance and decision support. Through analysis, it was found that the research on structural fault at home and abroad mostly focuses on structural fault monitoring and prediction, while the research on structural fault analysis based on system reliability theory is relatively lacking.

At present, scholars at home and abroad have carried out a series of studies on the reliability analysis of bridge structural systems. Since the structural system has many components and complex forms, which are difficult to calculate accurately, a series of more practical approximation methods have been derived, and these calculation methods can be roughly divided into two categories: "point estimation method" and "interval estimation method" [10].

Point estimation method mainly includes the Monte-Carlo method, direct numerical integration method, and approximate numerical calculation method [11]. The Monte-Carlo method mainly relies on computerized cyclic tests to respond to the response or performance of the structure as a whole by changing the parameters or design variables of the structural system. It is not only applicable to the reliability analysis of individual members, but also can approximate the failure of the system. However, since the accuracy of the Monte-Carlo method is based on a large amount of data, it is less applicable in the reliability analysis of large structural systems. The direct numerical integration method, as the name implies, solves the system reliability by directly calculating a multidimensional numerical integral. The probability of failure of a structural system composed of failure modes in the series becomes a problem of calculating a dimensional integral. However, the calculation process is very complicated and difficult to apply in practical engineering. Moreover, as the dimension of integration increases, the computational volume and error of integration grow exponentially, and the feasibility and accuracy are relatively low. The approximate numerical calculation method is used to convert the complex high-dimensional integration into a simple one-dimensional integration problem so as to solve the approximate solution with certain accuracy. The PENT method (Probabilistic Network Evaluation Technique) proposed by Ang H-S et al. is a typical approximate numerical computation method [11]. The basic principle of this method is that the failure modes are divided into several groups according to the closeness of their correlation and the failure mode with the highest failure probability in each group is selected as the representative failure mode, assuming that the representative failure modes are independent of each other. The failure probability of the system is estimated by using the following equation:

$$P_f = 1 - \prod_{i=1}^{m}\left(1 - P_{fi}\right) \tag{1}$$

where $P_f$ is the failure probability of the structural system; $m$ is the total number of representative failure modes; $i$ is the serial number of the representative failure modes, the value range of $i$ is $[1, m]$; and $P_{fi}$ is the failure probability of each representative failure mode. In addition to the above three "point estimation methods", many scholars have formulated a series of extensions and improvements on the basis of these methods to make them more relevant to practical engineering applications. Ditlevesen [12] proposed the reliability analysis method of series system by Taylor expansion of the cumulative distribution function of multidimensional normal distribution. However, for the reliability calculation of complex practical structural systems, the calculation amount is relatively large. M. Hohenbichler et al. proposed a first-order reliability calculation scheme by converting the non-normal correlation uncertainty vector into an independent standard normal vector to simplify the reliability calculation of tandem systems. Reference [13] proposed a conditional probability method based on the PENT method, which avoids the drawback of the PENT method.

The interval estimation method includes the wide and narrow bounds methods. Cornell [14] was the first to propose a wide-bound formula for the failure probability of a tandem system. The basic idea is that the failure probability of a structural system lies between the failure probability solved when the failure modes are completely correlated and the failure mode solved when the failure modes are completely independent. Since the wide bounds formula does not consider the correlation between the failure modes, it is generally used to roughly estimate the reliability of the structural system. To address the problem that the wide bounds method is too broad in scope, Ditleven [15] proposed a narrow-bounded range formula for the failure probability of a structural system. The narrow-bound method considers the probability of failure of two failure modes at the same time, which results in a narrower range than the wide-bound method and is more efficient in calculation. However, it only considers the failure of two failure modes together, and the estimation results are relatively rough. By analyzing domestic and foreign methods for calculating system reliability, it can be seen that the proposed system reliability calculation

methods have two major defects. Firstly, the calculation is too cumbersome to be applied to the actual complex engineering structures. Secondly, the correlation between failure modes is not considered comprehensively. Therefore, a method that is easy to calculate and accurately considers the correlation between failure modes is needed to guide the engineering practice.

In recent years, Copula theory has developed rapidly in the field of mathematics, providing a new way to establish the joint distribution function of related variables, which was first used in the field of finance. Copula theory was first proposed by Sklar [16] in the 1950s and Sklar's theorem laid the foundation for the application and development of Copula theory. Copula theory has been gradually used in the analysis of structural system reliability in recent years [17]. PL Liu et al. first applied the Copula function to structural reliability analysis, and pointed out that the essence of the Nataf transform is to use the Gaussian Copula function to construct the joint distribution function among variables [18]. Goda [19] used Copula theory to investigate the correlation between peak structural displacements and permanent displacements due to seismic loads. Kazianka and Pilz [20,21] used Copula function to describe the spatial dependence structure of continuous random field and discrete random field in geostatistics, provided three methods of spatial interpolation of geomechanical parameters based on Copula function, and, on this basis, proposed to consider the prior distribution of model parameters using Bayesian theory. Eryilmaz [22] achieved the time-varying reliability analysis of multivariate structural systems by studying multivariate random variable Copula functions. By considering the correlation of failure modes of structural systems, Yuefei Liu [23] proposed a reliability analysis method of structural systems based on the hybrid Copula model; the nonlinear correlation between the failure modes of binary and multivariate structural systems was reasonably analyzed, and the time-varying reliability of the structures was reasonably predicted with corrections based on the analysis results, which provides a new method for the reliability analysis of structural systems. Lei Zhang et al. [24] applied the Copula function into the reliability analysis of geotechnical structural systems and discussed the difference of failure probabilities of geotechnical structural systems calculated by different Copula functions, reflecting the necessity of making the optimal Copula function selection. Qingkai Xiao [25] applied Copula theory to the time-varying reliability analysis of bridge structures; the reliability of bridge structures during long-term operation was analyzed, and the reliability of bridge structures was predicted and analyzed based on Bayesian dynamic model, which provides a reference for the operation management of bridge projects. Xirui Wang [26], by applying Copula theory into the time-varying reliability analysis of existing small- and medium-span bridges, and proposing a bridge system reliability assessment method that considers multiple failure criteria based on the Copula function and AHP-EW (analytic hierarchy process–entropy weight) decision method, further improved the rationality of the bridge system reliability assessment. Laifu Song [27], using Copula theory to analyze the stability reliability of concrete dams, laid the foundation for the joint reliability analysis of engineering instances with few test data and where the joint distribution model cannot be determined. Copula theory is simple to calculate and easy to understand, and it is not limited by the amount of data. Copula theory is introduced into the reliability analysis of structural systems, which overcomes the problems of cumbersome calculation of traditional system reliability analysis methods and inadequate consideration of failure mode correlation. Although scholars at home and abroad have conducted a series of studies on the Copula function, the research on the Copula function is mainly focused on the theoretical level, and the research on structural reliability analysis based on Copula theory is still in the primary stage. Most of the existing structural system reliability analysis theories are focused on the bridge operation stage, and there are relatively few studies on the reliability of bridge system in the construction stage. Therefore, this paper proposed using Copula theory to analyze the reliability of the bridge system construction process.

Based on Copula function theory, this paper analyzed the reliability of structural systems during bridge construction. Four different Copula functions were used to establish

the correlation between failure modes. On this basis, the reliability index of the bridge structure system in the construction process was calculated with the help of MATLAB software. Then, the traditional Copula theory was improved according to the calculation results. Finally, the importance of failure mode correlation in practical engineering was explained through sensitivity analysis. This paper provides a technical reference for ensuring construction safety, improving construction quality, and controlling construction risks.

## 2. Basic Theory of the Copula Function

### 2.1. Definition of Copula Function

Copula theory was first proposed by Sklar [16]. Sklar's theory states that any multivariate joint distribution function can be decomposed into a corresponding marginal distribution and a Copula function, which uniquely determines the correlation between variables [28]. The Copula function is theoretically a joint distribution function, and Sklar's theorem is the basis for the practical application of the Copula function. According to Sklar's theorem, there exists only one unique Copula function satisfying the following Equation [29]:

$$F(x_1, x_2, \cdots x_n) = C(F_1(x_1), F_2(x_2), \cdots F_n(x_n); \theta) \tag{2}$$

where $C$ is the Copula function describing the structure of the correlation between the variables $(x_1, x_2, \cdots x_n)$, which is independent of the distribution with each marginal distribution $F_n(x_n)$; $\theta$ is the correlation parameter of the Copula function.

The correlation between the failure modes of the structural system is constructed on the basis of Copula theory in two main steps. Step 1 is to determine the failure function of each failure mode and to specify the distribution of each random variable in the failure function. Step 2 is to select the optimal Copula function to characterize the correlation structure between the failure modes. The two abovementioned steps are carried out independently without interfering with each other. It can be seen that the joint distribution function established when analyzing the reliability of the bridge construction period system based on Copula theory is not affected by the type and distribution of the failure function. The joint distribution function with any distribution type and correlation can be constructed in practical engineering. For the binary joint distribution, the joint cumulative distribution function $F(x_1, x_2)$ and the joint probability density function $f(x_1, x_2)$ are

$$F(x_1, x_2) = C(F_1(x_1), F_2(x_2); \theta) = C(u, v; \theta) \tag{3}$$

$$f(x_1, x_2) = D(u, v; \theta)f_1(x_1)f_2(x_2) \tag{4}$$

where $u = F_1(x_1)$ and $v = F_2(x_2)$ are the marginal cumulative distribution functions of the variables $X_1$ and $X_2$, respectively; $C(u, v; \theta)$ is the two-dimensional Copula function; $D(u, v; \theta)$ is the density function of the two-dimensional Copula function; $\theta$ is the correlation parameter of the Copula function, which is used to characterize the correlation between random variables.

### 2.2. Metrics of Copula Function Correlation

The correlation between the reliability of engineering structures can be measured by using multiple indicators; those commonly used include Pearson linear correlation coefficient, Kendall rank correlation coefficient, and Spearman rank correlation coefficient [28]. Since the Kendall rank correlation coefficient and Spearman rank correlation coefficient have similar properties, only the definition and calculation methods of Pearson linear correlation coefficient and Kendall rank correlation coefficient are introduced in this paper.

#### 2.2.1. Pearson Linear Correlation Coefficient

The Pearson linear correlation coefficient $\rho$ is a measure of the strength of the linear correlation between two random variables. We expand it to the field of reliability analysis. The Pearson linear correlation coefficient $\rho$ is calculated for the correlation between two functions as follows [10]

$$\rho_{Z_i Z_j} = \frac{\text{cov}(Z_i, Z_j)}{\sigma_{Z_i} \sigma_{Z_j}} \tag{5}$$

where $Z_i$ and $Z_j$ are the function of two failure modes, respectively; $\text{cov}(Z_i, Z_j)$ is the covariance of two failure functions $Z_i$ and $Z_j$; $\sigma_{Z_1}$ and $\sigma_{Z_2}$ are the standard deviation of two failure functions $Z_i$ and $Z_j$, respectively.

The Pearson linear correlation coefficient $\rho$ varies in the range of $[-1, 1]$. The larger the $|\rho|$, the stronger the correlation between the two failure modes. $|\rho| = 1$ indicates that the two failure modes are completely linearly correlated, and $|\rho| = 0$ indicates that there is no linear correlation between the two failure modes. Since the Pearson linear correlation coefficient can only describe the linear correlation between the failure modes, the Pearson linear correlation coefficient $\rho$ will be transformed when there is a nonlinear transformation between the failure modes [30]. Therefore, the applicability in engineering practice is weak.

### 2.2.2. Kendall Rank Correlation Coefficient

In contrast to the Pearson linear correlation coefficient, the Kendall rank correlation coefficient not only describes the linear correlation between failure modes, but also describes the nonlinear correlation between them well and has been widely used in engineering. The Kendall rank correlation coefficient is a measure of the degree of consistency between variables. Let $(x_{11}, x_{21})$ and $(x_{12}, x_{22})$ be two sets of observations of a two-dimensional random vector $(X_1, X_2)$. If $(x_{11} - x_{21})(x_{12} - x_{22}) > 0$, then $(x_{11}, x_{21})$ and $(x_{12}, x_{22})$ are said to be consistent. If $(x_{11} - x_{21})(x_{12} - x_{22}) < 0$, then $(x_{11}, x_{21})$ and $(x_{12}, x_{22})$ are said to be inconsistent. Establish the independent identically distributed vectors $(X_1', X_2')$ of the two-dimensional random vector $(X_1, X_2)$. Using $P[(X_1 - X_1')(X_2 - X_2') > 0]$ to denote their probability of being consistent, and $P[(X_1 - X_1')(X_2 - X_2') < 0]$ to denote their probability of being inconsistent, the Kendall rank correlation coefficient $\tau$ is defined as the difference between the probability of being consistent and the probability of being inconsistent, with the following equation [28]:

$$\tau = P[(X_1 - X_1')(X_2 - X_2') > 0] - P[(X_1 - X_1')(X_2 - X_2') < 0] \tag{6}$$

Similar to the range of the Pearson linear correlation coefficient $\rho$, the range of the Kendall rank correlation coefficient $\tau$ is also $[-1, 1]$. The larger the $|\tau|$, the stronger the correlation between the two failure modes. $|\tau| = 1$ indicates that the two failure modes are perfectly correlated, while $|\tau| = 0$ indicates that there is no correlation between the two failure modes. The Kendall rank correlation coefficient $\tau$ can also be calculated from the observations of the two-dimensional random vectors $X_1$ and $X_2$ [28]:

$$\tau = \frac{\sum\limits_{i<j} \text{sign}[(x_{1i} - x_{1j})(x_{2i} - x_{2j})]}{0.5N(N-1)} \tag{7}$$

where $i, j = 1, 2, \cdots, N$; $\text{sign}[\cdot]$ is the symbolic function, when $(x_{1i} - x_{1j})(x_{2i} - x_{2j}) > 0$, $\text{sign} = 1$, and vice versa, $\text{sign} = -1$; $N$ is the total number of combinations of observations.

The Kendall rank correlation coefficient $\tau$ has the following relationship with the two-dimensional Copula function $C(u, v; \theta)$ [28]:

$$\tau = 4 \int_0^1 \int_0^1 C(u, v; \theta) \text{d}C(u, v; \theta) - 1 \tag{8}$$

The Kendall rank correlation coefficient mainly reflects the correlation between two groups of random vectors and his independent identically distributed vectors. However, what this paper needs to reflect is the correlation between the two failure modes. In order to overcome the above drawbacks, this paper proposes to use the Latin hypercube sampling method to sample the uncertainty parameters in the two failure modes and then bring them into the failure mode function so as to calculate the response values of the function. The two sets of response values are the two-dimensional random vectors $X_1$ and $X_2$. Then the Kendall rank correlation coefficient $\tau$ between the two functions can then be

calculated using the Equation (6). The details of the Latin hypercube sampling method are described below.

The Latin hypercube sampling method to draw $N$ samples $x = (x_{1,i}, x_{2,i} \cdots x_{N,i})^T$ $(i = 1, 2 \cdots N)$ from the random variable $X = (X_1, X_2 \cdots X_m)^T$ is as follows [31]:

(1) Divide the range of each random variable $X_j$ into $N$ equal probability intervals, i.e., the domain of values of the cumulative distribution function $F_{X_j}(x_j)$ of the variable $X_j$ into $N$ equal subintervals $[0, 1/N], [1/N, 2/N] \ldots [1 - 1/N, 1]$ that do not overlap with each other.

(2) For each random variable $X_j$, a sample is taken in each of the $N$ subintervals divided in all its (1), and each subinterval generates a unique random number $x_{j,i}$ (indicating the sample random number drawn in the $i$-th interval of the $j$-th variable). The sample value corresponding to this random number is obtained by the inverse transformation method $\alpha_{j,i}$.

(3) The sample values of $\alpha_{j,i}$ are numbered 1,2,3 $\ldots$, $N$ from smallest to largest to form a matrix $A_{mn}$.

$$A_{mn} = \begin{bmatrix} \alpha_{11} & \alpha_{12} & \cdots & \alpha_{1n} \\ \alpha_{21} & \alpha_{22} & \cdots & \alpha_{2n} \\ \vdots & \vdots & \vdots & \vdots \\ \alpha_{m1} & \alpha_{m2} & \cdots & \alpha_{mn} \end{bmatrix} \tag{9}$$

The sample values of the m random variables from the first row to the $m$-th row are sorted from smallest to largest.

(4) Randomly sort the row vector $[\alpha_{j1}, \alpha_{j2} \cdots \alpha_{jn}] (j = 1, 2 \cdots m)$ in the matrix $A_{mn}$ and the resulting vector is denoted as $[\varphi_{j1}, \varphi_{j2} \cdots \varphi_{jn}] (j = 1, 2 \cdots m)$, resulting in the matrix $\Phi_{mn}$.

$$\Phi_{mn} = \begin{bmatrix} \varphi_{11} & \varphi_{12} & \cdots & \varphi_{1n} \\ \varphi_{21} & \varphi_{22} & \cdots & \varphi_{2n} \\ \vdots & \vdots & \vdots & \vdots \\ \varphi_{m1} & \varphi_{m2} & \cdots & \varphi_{mn} \end{bmatrix} \tag{10}$$

(5) Then, each column vector in $\Phi_{mn}$ is a set of samples, and a total of $N$ sets of samples are drawn.

### 2.3. Commonly Used Two-Dimensional Copula Functions

There are many types of Copula functions and the common ones include (1) two-dimensional elliptic Copula functions, such as Gaussian Copula and t Copula functions; (2) two-dimensional Plackett functions; and (3) two-dimensional Archimedean Copula functions, such as Frank, Clayton, No. 16 Copula functions, etc. In this paper, based on the review of the reference, the cumulative distribution function and probability density function of some commonly used Copula functions are listed at Table 1 [27].

### 2.4. Identification of Optimal Copula Functions

From the analysis of the current status of domestic and international research in the first section of this paper, it can be seen that the correlation between different failure modes is different, and different Copula functions are needed to describe the correlation between them. Therefore, how to accurately select the optimal Copula function to describe the correlation between failure modes is the primary problem that needs to be solved at present. There are many methods to identify the optimal Copula function, and there are two main ones used at present: (1) AIC (Akaike Information Criterion) [32]; and (2) BIC (Bayesian Information Criterion) [33].

**Table 1.** Commonly used two-dimensional Copula functions.

| Copula Function Type | Copula Function $C(u, v; \theta)$ | Copula Density Function $D(u, v; \theta)$ | Range Values of $\theta$ |
|---|---|---|---|
| Gaussian | $\Phi_\theta\big(\Phi^{-1}(u), \Phi^{-1}(v)\big)$ | $\frac{1}{\sqrt{1-\theta^2}} \exp\left[-\frac{\left(\Phi^{-1}(u)\right)^2\theta^2 - 2\theta\left(\Phi^{-1}(u)\cdot\Phi^{-1}(v)\right) + \left(\Phi^{-1}(v)\right)^2\theta^2}{2(1-\theta^2)}\right]$ | $[-1, 1]$ |
| Clayton | $\left(u^{-\theta} + v^{-\theta} - 1\right)^{-\frac{1}{\theta}}$ | $(1+\theta)(u\cdot v)^{-\theta-1}\left(u^{-\theta} + v^{-\theta} - 1\right)^{-2-1/\theta}$ | $(0, \infty)$ |
| Plackett | $\frac{S - \sqrt{S^2 - 4uv\theta(\theta-1)}}{2(\theta-1)}$ $S = 1 + (\theta-1)(u+v)$ | $\frac{\theta[1+(\theta-1)(u+v-2uv)]}{\left\{[1+(\theta-1)(u+v)]^2 - 4uv\theta(\theta-1)\right\}^{3/2}}$ | $(0, \infty)\backslash\{1\}$ |
| Frank | $-\frac{1}{\theta} \ln\left[1 + \frac{\left(e^{-\theta u}-1\right)\left(e^{-\theta v}-1\right)}{e^{-\theta}-1}\right]$ | $\frac{-\theta\left(e^{-\theta}-1\right)e^{-\theta(u+v)}}{\left[\left(e^{-\theta}-1\right)+\left(e^{-\theta u}-1\right)\left(e^{-\theta v}-1\right)\right]^2}$ | $(-\infty, \infty)\backslash\{0\}$ |
| $t$ | $T_2\big(T_v^{-1}(u_1), T_v^{-1}(u_2);\ \theta, v\big)$ | $\frac{t_2\left(T_v^{-1}(u_1), T_v^{-1}(u_2);\ \theta, v\right)}{t_v\left(T_v^{-1}(u_1)\right)t_v\left(T_v^{-1}(u_2)\right)}$ | $[-1, 1]$ |
| No. 16 | $\frac{1}{2}\left(S + \sqrt{S^2 + 4\theta}\right)$ $S = u + v - 1 - \theta\left(\frac{1}{u} + \frac{1}{v} - 1\right)$ | $\frac{1}{2}\left(1 + \frac{\theta}{u_1^2}\right)\left(1 + \frac{\theta}{u_2^2}\right)S^{-\frac{1}{2}}\left\{-S^{-1}\left[u+v-1-\theta\left(\frac{1}{u}+\frac{1}{v}-1\right)\right]^2 + 1\right\}$ $S = \left[u+v-1-\theta\left(\frac{1}{u}+\frac{1}{v}-1\right)\right]^2 + 4\theta$ | $[0, \infty)$ |
| Gumbel | $\exp\left\{-\left[(-\ln u_1)^\theta + (-\ln u_2)^\theta\right]^{1/\theta}\right\}$ | $\frac{e^{(-S^{1/\theta})}(\ln u_1 \ln u_2)^{\theta-1}\left(S^{1/\theta} + \theta + 1\right)}{u_1 u_2 S^{2-1/\theta}}$ $S = (-\ln u_1)^\theta + (-\ln u_2)^\theta$ | $[1, \infty)$ |

AIC and BIC are based on the principle that the Copula function with the smallest AIC or BIC value is considered to be the optimal Copula function for fitting the correlation between the two data sets. AIC is based on the concept of entropy, which is a standard to measure the goodness of statistical model fitting. The specific method is to find a model that contains the least free parameters and can best interpret the data. The AIC value is defined as the sum of −2 times the logarithmic sum of the Copula density function values at the original observed data points of the variable and 2 times the number of parameters associated with the Copula function [32]. When there is a large difference between the two models, the difference is mainly reflected in the likelihood function term. When the likelihood function difference is not significant, the model complexity plays a role, so the model with less parameters is a better choice. Generally, when the complexity of the model increases, the likelihood function will also increase so that the AIC becomes smaller. However, when the complexity of the model is too large, the growth rate of the likelihood function slows down, resulting in the increase of AIC. If the model is too complex, it is easy to cause over fitting. The model selection problem seeks the best balance between the complexity of the model and the ability of the model to describe the data set (i.e., likelihood function). BIC is similar to AIC for model selection. The BIC value is defined as the sum of −2 times the logarithmic sum of the Copula density function values at the original observed data points and $\ln N$ times the number of parameters associated with the Copula function [33]. The first half of AIC and BIC calculation formula is the same, and the second half is the penalty term. BIC penalizes model parameters more than AIC when there is a large amount of data, resulting in BIC preferring to choose simple models with fewer parameters. The equations for the calculation of AIC and BIC are as follows:

$$\text{AIC} = -2\sum_{i=1}^{N} \ln D(u_i, v_i;\ \theta) + 2n \tag{11}$$

$$\text{BIC} = -2\sum_{i=1}^{N} \ln D(u_i, v_i;\ \theta) + 2n \ln N \tag{12}$$

where $n$ is the number of relevant parameters in the Copula function, for a two-dimensional Copula function $n = 1$; $(u_i, v_i)$ is the empirical distribution value of the original observation $(x_{1i}, x_{2i})$, which can be calculated by the following equation:

$$\begin{cases} u_i = \frac{rank(x_i)}{N+1} \\ v_i = \frac{rank(y_i)}{N+1} \end{cases}, i = 1, 2, \cdots, N \tag{13}$$

where $rank(\cdot)$ denotes the rank of samples $x_i$ or $y_i$, i.e., the number of samples arranged from smallest to largest is less than or equal to $x_i$ or $y_i$.

In the reliability analysis process, the observed data of the failure modes cannot be known, but a method to establish a random vector of two failure modes using the Latin hypercube sampling method is proposed in Section 2.2.2 of this paper. This two-dimensional random vector can be brought into Equations (11)–(13) when calculating the correlation between the failure modes to expand the calculation. We found the AIC and BIC values of the alternative Copula functions, which could be compared to identify the optimal Copula function suitable for the correlation of these two failure modes.

## 3. Engineering Applications

In order to more accurately and intuitively reflect the superiority of the structural system reliability calculation method based on Copula theory during the bridge construction period, this paper used the calculated engineering examples in reference [34], and compared them with the traditional system reliability calculation method.

### 3.1. Basic Information of the Algorithm

Reference [34] analyzed the reliability of the long-span continuous rigid frame bridge system based on the engineering background of the Labajin Super Large Bridge in the Yingjing section of Yahu Expressway in China. The main span of the bridge is a separated variable cross-section continuous rigid frame, and the span combination of the whole bridge is (105 + 2 × 200 + 105) m. The main span is 610 m long, the main beam of the superstructure adopts a prestressed single-box single chamber section, and the bridge is a fully prestressed concrete structure. In order to reduce the amount of calculation, reference [34] only took the single span bridge of the main span for calculation. The steps of analyzing the reliability of bridge system in reference [34] can be roughly divided into the following four steps.

(1) Use MIDAS/Civil software to establish the finite element analysis model of the whole bridge.
(2) Use the modified $\beta$-bound method to search for the main failure modes of the bridge structure.
(3) Use the improved quadratic series response surface method to solve the failure function in different failure modes.
(4) Use the wide boundary method, narrow boundary method, and PENT method to calculate the reliability indexes of the continuous rigid bridge system in the construction stage respectively.

The basic information related to the calculations in this paper in reference [34] is shown below.

In reference [34], the entire bridge construction process is divided into 28 construction stages. Each construction stage includes four processes: moving hanging baskets, tying reinforcement, placing concrete, and tensioning prestressing. The maximum cantilever construction stage of the middle pier "T" structure was selected as an example for the system reliability analysis. For the mechanical characteristics of this continuous rigid bridge, three failure modes were selected: axial compression stability failure and tensile and compressive stress failure during the construction and operation periods. The failure of any section of the main girder or the instability of any pier will lead to the failure of the bridge system, so each failure mode forms a tandem system. The FEA results were used to find the location where the stress in the main girder section is relatively large or small and the tensile and compressive stress failure of the main girder at this location and the stability failure of pier 10 were selected as the main failure modes as shown in the Table 2 below.

**Table 2.** Basic information of each failure mode.

| Failure Mode Number | 1 | 2 | 3 | 4 | 5 | 6 | 7 |
|---|---|---|---|---|---|---|---|
| Location/Node | Pier 10 | 65 | 79 | 90 | 96 | 107 | 121 |
| Failure mode | Stability | Lower edge Tensile stress | Lower edge Compressive stress | Lower edge Compressive stress | Lower edge Compressive stress | Lower edge Compressive stress | Lower edge Tensile stress |

The functions of each failure mode in the construction phase were obtained by fitting the modified quadratic series response surface method as follows.

$$g_1(X) = 430.966K_{R1} - 15.389K_{R1}K_G - 101.534K_G \tag{14}$$

$$g_2(X) = 1.96K_{R3} - (0.74 - 0.28K_G + 0.29K_E + 0.078K_\xi - 0.12K_{Q1} \\ -0.13K_E^2 + 0.2K_\xi^2 + 0.01K_{Q1}^2) \tag{15}$$

$$g_3(X) = 26.5K_{R2} - (-10.66 + 19.52K_G + 4.11K_E + 5.44K_\xi + 1.11K_{Q1} \\ -2.49K_G^2 - 2.22K_E^2 - 2.22K_\xi^2 + 0.27K_{Q1}^2) \tag{16}$$

$$g_4(X) = 26.5K_{R2} - (-0.41 + 12.67K_G - 4.78K_E - 2.78K_\xi - 0.19K_{Q1} \\ +0.99K_G^2 + 2.22K_E^2 - 2.22K_\xi^2 + 0.27K_{Q1}^2) \tag{17}$$

$$g_5(X) = 26.5K_{R2} - (-5.07 + 14.83K_G + 1.33K_\xi + 0.23K_{Q1}) \tag{18}$$

$$g_6(X) = 26.5K_{R2} - (6.15 - 0.66K_G - 4.11K_E + 0.67K_\xi + 1.35K_{Q1} \\ +7.48K_G^2 + 2.22K_E^2 - 0.27K_{Q1}^2) \tag{19}$$

$$g_7(X) = 26.5K_{R2} - (1.15 + 8.72K_G - 0.67K_E + 0.67K_\xi + 0.46K_{Q1} + 2.49K_G^2) \tag{20}$$

where $K_{R1}$ is the uncertainty factor for stability resistance calculations; $K_{R2}$ is the axial compressive strength uncertainty factor for C60 concrete for the main beam; $K_{R3}$ is the axial tensile strength uncertainty factor of C60 concrete for the main beam; $K_G$ is the weight uncertainty of the member, $K_G = \frac{G}{G_K}$ ($G$ is the actual member weight, $G_K$ is the standard weight of the member specified in the current code); $K_E$ is the equivalent modulus of elasticity uncertainty factor; $K_\xi$ is the stress pipe friction uncertainty factor; $K_{Q1}$ is the construction period expressed as the construction load uncertainty factor. These uncertainty factors obey the distribution functions and characteristic parameters taken as the following Table 3.

**Table 3.** Basic information of each uncertainty factor.

| Name | $K_{R1}$ | $K_{R2}$ | $K_{R3}$ | $K_G$ | $K_E$ | $K_\xi$ | $K_{Q1}$ |
|---|---|---|---|---|---|---|---|
| Distribution | Normal | Normal | Normal | Normal | Normal | Normal | Extreme vale type I |
| $\mu$ | 1.3636 | 1.3849 | 1.1420 | 1.0212 | 1.0000 | 1.0000 | 1.0000 |
| $\delta$ | 0.1344 | 0.1406 | 0.0967 | 0.0462 | 0.0500 | 0.1440 | 0.0862 |

Using the reliability calculator of reference [34] can find the reliability indicators for each failure mode as follows: $\beta_1 = 7.93$, $\beta_2 = 8.77$, $\beta_3 = 4.62$, $\beta_4 = 5.02$, $\beta_5 = 4.81$, $\beta_6 = 4.52$, $\beta_7 = 8.36$. The probability of failure for each of the seven different reliability indicators was calculated using MATLAB. Since the reliability indicator of $\beta_2$ is very large, the probability of failure after expansion is very small and can be approximated as 0. The results of their respective probability of failure calculations are as follows: $P_{f_1} = 1.1102 \times 10^{-15}, P_{f_2} = 0, P_{f_3} = 1.9187 \times 10^{-6}, P_{f_4} = 2.5836 \times 10^{-7}, P_{f_5} = 7.5465 \times 10^{-7}, P_{f_6} = 3.0920 \times 10^{-6}, P_{f_7} = 5.5511 \times 10^{-7}$.

*3.2. Calculation of System Reliability Indicators*

3.2.1. Establishment of Copula Function

Analyzing the basic case of the calculation in Section 3.1, it can be seen that the failure of any one mode in the construction process will lead to the failure of the whole structural system. So, the failure modes are in series with each other. The joint failure probability of the bridge structure system construction process failure occurring is shown in the following equation. For simplifying the calculation, this paper only considered the correlation between two failure modes.

$$
\begin{aligned}
P_f &= P\{Z_1(X) \le 0 \cup Z_2(X) \le 0 \cup \cdots \cup Z_n(X) \le 0\} \\
&= \sum_{i=1}^{n} P\{Z_i(X) \le 0\} - \sum_{1 \le i < j \le n} P\{Z_i(X) \le 0, Z_j(X) \le 0\} \\
&\quad + \sum_{1 \le i < j < k \le n} P\{Z_i(X) \le 0, Z_j(X) \le 0, Z_k(X) \le 0\} \\
&\quad + \cdots + (-1)^{n-1} P\{Z_1(X) \le 0, Z_2(X) \le 0, \cdots, Z_n(X) \le 0\} \\
&\approx \sum_{i=1}^{n} P\{Z_i(X) \le 0\} - \sum_{1 \le i < j \le n} P\{Z_i(X) \le 0, Z_j(X) \le 0\}
\end{aligned}
\tag{21}
$$

Bringing in the arithmetic example in Section 3.1, Equation (21) becomes

$$
\begin{aligned}
P_f &= P\{g_1(X) \le 0 \cup g_2(X) \le 0 \cup g_3(X) \le 0 \cup g_4(X) \le 0 \cup \\
&\quad g_5(X) \le 0 \cup g_6(X) \le 0 \cup g_7(X) \le 0\}
\end{aligned}
\tag{22}
$$

The correlation between failure modes was established using the Copula function, and according to Equation (2), it can be seen that the bridge system failure probability is a 7-dimensional Copula function. According to Equation (3), it can be derived that the probability of failure mode occurrence of the binary combined structural system is shown in the following equation.

$$
P(g_1(X) \le 0, g_2(X) \le 0) = C\left(P_{f_1}, P_{f_2}; \theta_{12}\right)
\tag{23}
$$

Treating each failure mode as a probability event. According to the addition formula of multiple probability events in probability theory, the corresponding calculation formula of the failure probability of the bridge system is obtained, in which the joint failure probability of failure criterion is calculated according to Copula theory. Then Equation (22) can be converted into the following equation.

$$
\begin{aligned}
P_f &= \{g_1(X) \le 0 \cup g_2(X) \le 0 \cup g_3(X) \le 0 \cup g_4(X) \le 0 \cup \\
&\quad g_5(X) \le 0 \cup g_6(X) \le 0 \cup g_7(X) \le 0 \\
&= P\{g_1(X) \le 0\} + P\{g_2(X) \le 0\} + P\{g_3(X) \le 0\} \\
&\quad + P\{g_4(X) \le 0\} + P\{g_5(X) \le 0\} + P\{g_6(X) \le 0\} \\
&\quad + P\{g_7(X) \le 0\} - C\left(P_{f_1}, P_{f_2}\right) - C\left(P_{f_1}, P_{f_3}\right) - C\left(P_{f_1}, P_{f_4}\right) \\
&\quad - C\left(P_{f_1}, P_{f_5}\right) - C\left(P_{f_1}, P_{f_6}\right) - C\left(P_{f_1}, P_{f_7}\right) - C\left(P_{f_2}, P_{f_3}\right) \\
&\quad - C\left(P_{f_2}, P_{f_4}\right) - C\left(P_{f_2}, P_{f_5}\right) - C\left(P_{f_2}, P_{f_6}\right) - C\left(P_{f_2}, P_{f_7}\right) \\
&\quad - C\left(P_{f_3}, P_{f_4}\right) - C\left(P_{f_3}, P_{f_5}\right) - C\left(P_{f_3}, P_{f_6}\right) - C\left(P_{f_3}, P_{f_7}\right) \\
&\quad - C\left(P_{f_4}, P_{f_5}\right) - C\left(P_{f_4}, P_{f_6}\right) - C\left(P_{f_4}, P_{f_7}\right) - C\left(P_{f_5}, P_{f_6}\right) \\
&\quad - C\left(P_{f_5}, P_{f_7}\right) - C\left(P_{f_6}, P_{f_7}\right)
\end{aligned}
\tag{24}
$$

3.2.2. Calculation of the Kendall Rank Correlation Coefficient $\tau$

According to the contents described in Section 2.2.2 of this paper, the calculation of the Kendall rank correlation coefficient $\tau$ between failure modes requires a Latin hypercube sampling of the seven uncertainty factors (Table 3). In this paper, we used MATLAB software to divide the defined domain of each random variable into 500 groups and

generated a random datum in each group, after which we disrupted its order to form a matrix $\Phi_{mn}$. Since this sample is a huge matrix of $500 \times 7$, it is not listed in detail here. Afterwards, the samples of each random variable were brought into the function of each failure mode, using Equation (7) to solve the Kendall rank correlation coefficients between the two functions. The Kendall rank correlation coefficients for the 21 sets of failure mode combinations were solved as shown in the following Table 4.

**Table 4.** Calculations of Kendall rank correlation coefficients for each combination of failure modes.

| $g_i g_j$ | $\tau$ | $g_i g_j$ | $\tau$ | $g_i g_j$ | $\tau$ |
|---|---|---|---|---|---|
| $g_1 g_2$ | 0.00983 | $g_2 g_4$ | 0.00386 | $g_3 g_7$ | 0.97839 |
| $g_1 g_3$ | −0.01303 | $g_2 g_5$ | 0.01034 | $g_4 g_5$ | 0.94616 |
| $g_1 g_4$ | −0.01323 | $g_2 g_6$ | 0.00915 | $g_4 g_6$ | 0.94947 |
| $g_1 g_5$ | −0.01242 | $g_2 g_7$ | 0.01015 | $g_4 g_7$ | 0.95101 |
| $g_1 g_6$ | −0.01259 | $g_3 g_4$ | 0.94389 | $g_5 g_6$ | 0.98746 |
| $g_1 g_7$ | −0.01345 | $g_3 g_5$ | 0.97422 | $g_5 g_7$ | 0.99070 |
| $g_2 g_3$ | 0.00960 | $g_3 g_6$ | 0.98442 | $g_6 g_7$ | 0.98955 |

From the above table, it can be seen that the correlation between 1, 2, and other failure modes was very low. As mentioned in Section 2.2.2 of this paper, the range of the Kendall rank correlation coefficient $\tau$ is $[-1, 1]$. The larger the $|\tau|$, the stronger the correlation between the two failure modes. $|\tau| = 1$ indicates that the two failure modes are perfectly correlated, while $|\tau| = 0$ indicates that there is no correlation between the two failure modes. Since the correlation coefficients between 1, 2, and other failure modes were all around 0, there was almost no linear correlation between the two failure modes. By consulting the literature, when $|\tau|$ is greater than 0.5, the correlation between failure modes should not be ignored. To simplify the calculation, the correlation between 1, 2, and other failure modes is no longer considered here, so Equation (24) is changed to:

$$
\begin{aligned}
P_f &= P\{g_1(X) \le 0, g_2(X) \le 0, g_3(X) \le 0, g_4(X) \le 0, \\
&\quad g_5(X) \le 0, g_6(X) \le 0, g_7(X) \le 0\} \\
&= P\{g_1(X) \le 0\} + P\{g_2(X) \le 0\} + P\{g_3(X) \le 0\} \\
&\quad + P\{g_4(X) \le 0\} + P\{g_5(X) \le 0\} + P\{g_6(X) \le 0\} \\
&\quad + P\{g_7(X) \le 0\} - C\left(P_{f_3}, P_{f_4}\right) - C\left(P_{f_3}, P_{f_5}\right) - C\left(P_{f_3}, P_{f_6}\right) \\
&\quad - C\left(P_{f_3}, P_{f_7}\right) - C\left(P_{f_4}, P_{f_5}\right) - C\left(P_{f_4}, P_{f_6}\right) - C\left(P_{f_4}, P_{f_7}\right) \\
&\quad - C\left(P_{f_5}, P_{f_6}\right) - C\left(P_{f_5}, P_{f_7}\right) - C\left(P_{f_6}, P_{f_7}\right)
\end{aligned}
\tag{25}
$$

### 3.2.3. Selection of Copula Function and Calculation of Related Parameters

Based on the review of a large amount of references, four Copula functions commonly used in engineering were selected in this paper to carry out the analysis of failure mode correlation. These four functions are Gaussian Copula function, Clayton Copula function, Frank Copula function, and Gumbel Copula function.

We used Equation (8) to calculate the values of the relevant parameters for each failure mode in the case of four different combinations of Copula functions separately, and the results are listed at Table 5.

### 3.2.4. Identification of Optimal Copula Functions

In this paper, we used the AIC and the BIC for the selection of the optimal Copula function. According to Equations (11)–(13), the AIC and BIC values were calculated for each combination of different failure modes when fitted with each of the four Copula functions, and the results are shown in the following Table 6.

**Table 5.** Relevant parameter values of the alternative Copula function.

| $g_i g_j$ | Copula Function $\theta$ | | | |
| | Gaussian | Clayton | Frank | Gumbel |
|---|---|---|---|---|
| $g_3 g_4$ | 0.99610 | 33.6443 | 1.3755 | 17.8221 |
| $g_3 g_5$ | 0.99920 | 75.5795 | 1.4054 | 38.7898 |
| $g_3 g_6$ | 0.99970 | 126.3697 | 1.4163 | 64.1849 |
| $g_3 g_7$ | 0.99940 | 90.5497 | 1.4098 | 46.2749 |
| $g_4 g_5$ | 0.99640 | 35.1471 | 1.3776 | 18.5736 |
| $g_4 g_6$ | 0.99690 | 37.5804 | 1.3808 | 19.7902 |
| $g_4 g_7$ | 0.99700 | 38.8247 | 1.3823 | 20.4123 |
| $g_5 g_6$ | 0.99980 | 157.4896 | 1.4196 | 79.7448 |
| $g_5 g_7$ | 0.99989 | 213.0538 | 1.4231 | 107.5269 |
| $g_6 g_7$ | 0.99986 | 189.3876 | 1.4218 | 95.6938 |

**Table 6.** List of calculated results of AIC and BIC values (the bolded parts in the table are the minimum values of AIC and BIC for each failure mode combination).

| $g_i g_j$ | Gaussian | | Clayton | | Frank | | Gumbel | |
| | AIC | BIC | AIC | BIC | AIC | BIC | AIC | BIC |
|---|---|---|---|---|---|---|---|---|
| $g_3 g_4$ | −2394.7 | −2390.5 | −1848.5 | −1844.3 | −211.3 | −207.1 | **−2431.1** | **−2426.9** |
| $g_3 g_5$ | **−3092.1** | **−3087.9** | −2309.7 | −2305.5 | −216.7 | −212.4 | −3064.3 | −3060.1 |
| $g_3 g_6$ | **−3493.5** | **−3489.2** | −2693.9 | −2689.7 | −218.4 | −214.2 | −3443.5 | −3439.3 |
| $g_3 g_7$ | **−3255.7** | **−3251.5** | −2566.8 | −2562.5 | −217.4 | −213.2 | −3186.6 | −3182.3 |
| $g_4 g_5$ | −2444.2 | −2440.0 | −1871.0 | −1866.8 | −211.7 | −207.5 | **−2502.7** | **−2498.5** |
| $g_4 g_6$ | −2507.3 | −2503.0 | −1941.4 | −1937.2 | −212.3 | −208.1 | **−2550.0** | **−2545.8** |
| $g_4 g_7$ | −2531.2 | −2526.9 | −1962.6 | −1958.4 | −212.6 | −208.4 | **−2566.7** | **−2562.5** |
| $g_5 g_6$ | **−3723.4** | **−3719.2** | −2857.7 | −2853.5 | −218.9 | −214.7 | −3614.9 | −3610.7 |
| $g_5 g_7$ | **−3901.5** | **−3897.3** | −2816.5 | −2812.3 | −219.5 | −215.3 | −3759.5 | −3755.3 |
| $g_6 g_7$ | **−3875.4** | **−3871.2** | −2829.3 | −2825.1 | −219.3 | −215.1 | −3667.6 | −3663.4 |

According to the calculation results, it can be seen that the failure mode combinations $g_3 g_4$, $g_4 g_5$, $g_4 g_6$, and $g_4 g_7$ should use the Gumbel Copula function to establish the binary joint distribution, and the failure mode combinations $g_3 g_5$, $g_3 g_6$, $g_3 g_7$, $g_5 g_6$, $g_5 g_7$, and $g_6 g_7$ should use the Gaussian Copula function to establish the binary joint distribution.

3.2.5. Calculation of the System Failure Probability

Based on the results of the analysis in Section 3.2.4, the failure probabilities of the failure mode combinations are brought into the corresponding Copula functions, respectively, and the values of the joint failure probabilities of the two-dimensional failure modes can be calculated as shown in the following Table 7.

**Table 7.** Two-dimensional joint failure probabilities for each combination of failure modes.

| $g_i g_j$ | Joint Failure Probability | $g_i g_j$ | Joint Failure Probability |
|---|---|---|---|
| $g_3 g_4$ | $2.4196 \times 10^{-7}$ | $g_4 g_6$ | $2.5273 \times 10^{-7}$ |
| $g_3 g_5$ | $7.5465 \times 10^{-7}$ | $g_4 g_7$ | $5.5511 \times 10^{-17}$ |
| $g_3 g_6$ | $1.9187 \times 10^{-6}$ | $g_5 g_6$ | $7.5465 \times 10^{-7}$ |
| $g_3 g_7$ | $5.5511 \times 10^{-17}$ | $g_5 g_7$ | $5.5511 \times 10^{-17}$ |
| $g_4 g_5$ | $2.1418 \times 10^{-7}$ | $g_6 g_7$ | $5.5511 \times 10^{-17}$ |

From the failure probability values calculated in the above table, it can be seen that all the two-dimensional joint failure probabilities combined with the seventh failure mode have a large order of magnitude, all of which were $10^{-17}$. It can be concluded that the correlation between the seventh failure mode and the other four failure modes is weak. The orders of magnitude of the failure probability values calculated by the other four failure

modes were similar, which were all around $10^{-6}$ and $10^{-7}$. It can be seen that these four failure modes are highly correlated. Therefore, the influence of failure mode correlation on the reliability of the structural system should not be ignored in practical engineering.

Bringing the individual two-dimensional joint failure probabilities into Equation (25), the probability of failure of the bridge structure system during the construction period can be calculated as $1.8869 \times 10^{-6}$, and the system reliability index is 4.6235.

According to Section 5.5 of reference [34], the reliability index range calculated by the traditional wide bound method was (4.376, 4.52), and the reliability index range calculated by the narrow limit method was (4.4177, 4.4455), and the reliability index calculated by PENT method was 4.52. Compared with the reliability index 4.6235 calculated in this paper, it can be seen that the reliability index of the bridge structure system calculated in this paper is relatively high. Based on the additional calculation and analysis of other reference examples, it is concluded that the reliability of the bridge construction period system based on Copula theory is safer.

3.2.6. Improvement of Computational Theory

In order to better guide the actual bridge construction project, this paper makes the following improvements to the traditional Copula theory.

According to the calculation process in Section 3.2.5 of this paper, when establishing the linkage between failure modes, the correlation between two of the individual failure modes is established many times. So, when applying Equation (25), the same indicator is subtracted from other indicators several times when calculating. For example, in $C\left(P_{f_3}, P_{f_4}\right)$, $C\left(P_{f_3}, P_{f_5}\right)$, $C\left(P_{f_3}, P_{f_6}\right)$, and $C\left(P_{f_3}, P_{f_7}\right)$, the linkage between failure mode 3 and other failure modes is established several times, and in $C\left(P_{f_3}, P_{f_6}\right)$, $C\left(P_{f_4}, P_{f_6}\right)$, $C\left(P_{f_5}, P_{f_6}\right)$, and $C\left(P_{f_6}, P_{f_7}\right)$, the linkage between failure mode 6 and other failure modes is established several times. Although the links established are between two different failure modes, if we directly use Equation (25), the probability of partial failure of the same failure mode will be subtracted several times, which will lead to a low failure probability, large reliability index, and safe calculation results. Therefore, this paper proposes the following improvement: when subtracting the Copula joint distribution of the correlation between failure modes, the inverse of the number of occurrences of such failure modes should be multiplied as a coefficient before the Copula function. According to Equations (21) and (23), the improved equation is shown in the following equation.

$$
\begin{aligned}
P_f \; &= P\{Z_1(X) \le 0 \cup Z_2(X) \le 0 \cup \cdots \cup Z_n(X) \le 0\} \\
&= \sum_{i=1}^{n} P\{Z_i(X) \le 0\} - \sum_{1 \le i < j \le n}^{n} P\{Z_i(X) \le 0, Z_j(X) \le 0\} \\
&\quad + \sum_{1 \le i < j < k \le n}^{n} P\{Z_i(X) \le 0, Z_j(X) \le 0, Z_k(X) \le 0\} \\
&\quad + \cdots + (-1)^{n-1} P\{Z_1(X) \le 0, Z_2(X) \le 0, \cdots, Z_n(X) \le 0\} \\
&\approx \sum_{i=1}^{n} P\{Z_i(X) \le 0\} - \sum_{1 \le i < j \le n}^{n} P\{Z_i(X) \le 0, Z_j(X) \le 0\} \\
&= \sum_{i=1}^{n} P_{f_{Z_i}} - \frac{1}{j_{\max} - i_{\min}} \sum_{1 \le i < j < n}^{n} C\left(P_{f_{Z_i}}, P_{f_{Z_j}}\right)
\end{aligned}
\tag{26}
$$

We recalculated the reliability index of the bridge system in the construction process in the example of this paper according to Equation (26). Equation (25) becomes the following equation.

$$
\begin{aligned}
P_f \quad &= P\{g_1(X) \le 0, g_2(X) \le 0, g_3(X) \le 0, g_4(X) \le 0, \\
&\quad g_5(X) \le 0, g_6(X) \le 0, g_7(X) \le 0\} \\
&= P\{g_1(X) \le 0\} + P\{g_2(X) \le 0\} + P\{g_3(X) \le 0\} \\
&\quad + P\{g_4(X) \le 0\} + P\{g_5(X) \le 0\} + P\{g_6(X) \le 0\} \\
&\quad + P\{g_7(X) \le 0\} - \tfrac{1}{4}[C\left(P_{f_3}, P_{f_4}\right) - C\left(P_{f_3}, P_{f_5}\right) - C\left(P_{f_3}, P_{f_6}\right) \\
&\quad - C\left(P_{f_3}, P_{f_7}\right) - C\left(P_{f_4}, P_{f_5}\right) - C\left(P_{f_4}, P_{f_6}\right) - C\left(P_{f_4}, P_{f_7}\right) \\
&\quad - C\left(P_{f_5}, P_{f_6}\right) - C\left(P_{f_5}, P_{f_7}\right) - C\left(P_{f_6}, P_{f_7}\right)]
\end{aligned}
\tag{27}
$$

Using Equation (27), it was calculated that the failure probability of the system during the construction of the bridge structure was $4.9895 \times 10^{-6}$. With the help of MATLAB software, the system reliability index aw 4.4176. Compared with the calculation results in Section 3.2.5 of this paper, it was found that the improved Copula theory for calculating the reliability of bridge structure system overcomes the problem that the calculation results of the traditional copula theory are partial to safety. The calculation results fell within the boundary range (4.376, 4.52) calculated by the wide boundary method.

*3.3. System Reliability Analysis*

3.3.1. Relationship between System Reliability Index and Failure Mode Reliability Index

In this section, the relationship between the variation of the reliability indexes of the bridge system with the reliability indexes of individual failure modes is analyzed, and the system reliability index values are calculated separately when each single failure mode takes any integer between 1 and 8. Based on these data, a graph of the variation of the system reliability indexes with each single reliability index is fitted and the variation is plotted in Figure 1. From Figure 1, it can be seen that the seven failure modes mentioned in the examples used in this paper have approximately the same effect on the system reliability index of the bridge structure. When the reliability index of a single failure mode is less than 4, the reliability index of a single failure mode directly determines the reliability index of the whole system due to the high probability of failure. As the individual failure mode reliability index increases, the reliability index of the bridge system tends to be stable and finally stabilizes at a number near the system reliability index calculated in this paper (4.4176). The reading of the graph shows that the sixth failure mode, the compressive stress at the lower edge of node 107, plays a decisive role in the magnitude of the system reliability index, which is consistent with the definition used in the wide boundary method calculation.

In order to more clearly describe the influence of each reliability index of the overlapping part in Figure 1 on the system reliability index, the influence of a single failure mode reliability index on the system reliability index when the variation range of the reliability index is 4~8 is enlarged in Figure 2.

3.3.2. Variation of System Failure Probability with Failure Mode Correlation Coefficient

This section analyzes the relationship between the reliability index of the bridge structural system and the Kendall rank correlation coefficient of the failure mode combination. The reliability index value of the system was calculated when the Kendall rank correlation coefficient of each failure mode combination was taken as 19 key data points between 0–1. Based on these data, the curves of the system reliability indexes with the change of the Kendall rank correlation coefficient for each failure mode combination were fitted as shown in Figure 3. From Figure 3, it can be seen that the Kendall rank correlation coefficients of the 3rd failure mode and the 6th failure mode combination had a relatively great influence on the reliability indexes of the bridge structure system; the combination of failure mode 3 with failure mode 5 and the combination of failure mode 5 with failure mode 6 had a relatively greater influence on the reliability indexes of the bridge structure system; the combination of failure mode 4 with failure mode 5, failure mode 4 with failure mode 6, and failure mode 3 with failure mode 4 had relatively less influence on the reliability index of

the bridge structure system; all the combinations associated with failure mode 7 had little influence on the reliability index of the structure system. Reference [34] points out that the main failure mode of the maximum cantilever state during construction is the lower edge compressive stress at 1/4 of the bridge span. Failure mode 3 and failure mode 6 are located at 1/4 of the bridge span. Therefore, the combination of failure modes 3 and 6 had a great impact on the reliability of structural system. The influence of failure mode combination on the failure probability of structural system is mainly reflected in the level of single-failure probability. The higher the failure probability of the failure mode, the greater the influence of this failure mode combined with other failure modes on the reliability of the structural system. This is consistent with the meaning indicated by the curves in Figure 3.

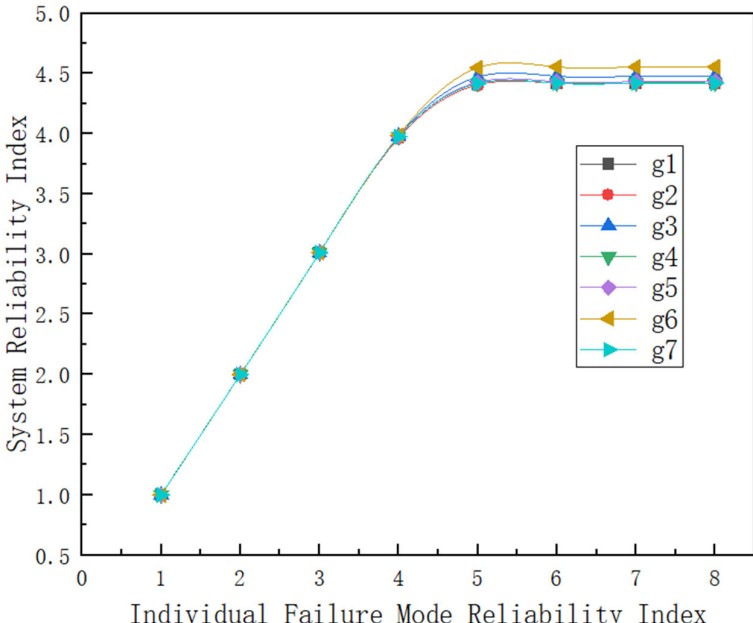

**Figure 1.** Variation of system reliability indexes with individual failure mode reliability index.

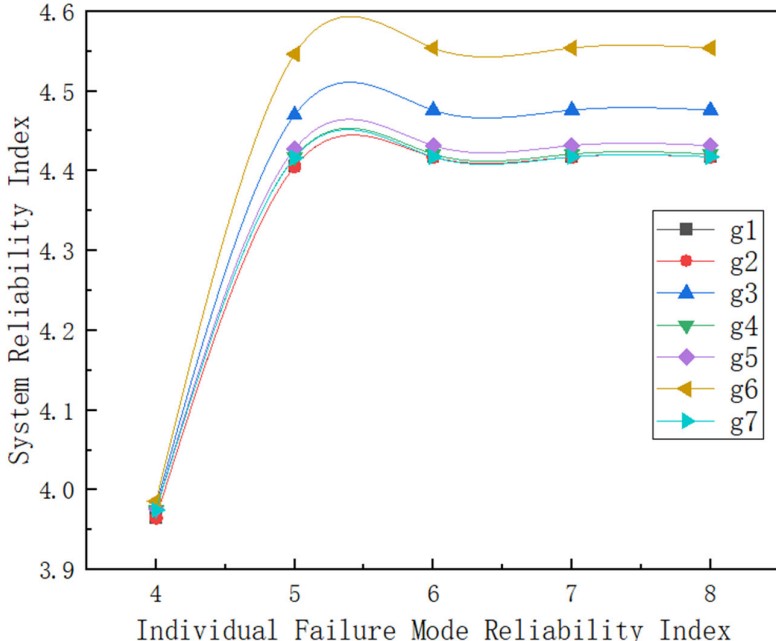

**Figure 2.** Variation diagram of system reliability index with single failure mode reliability index when the variation range of reliability index is 4~8.

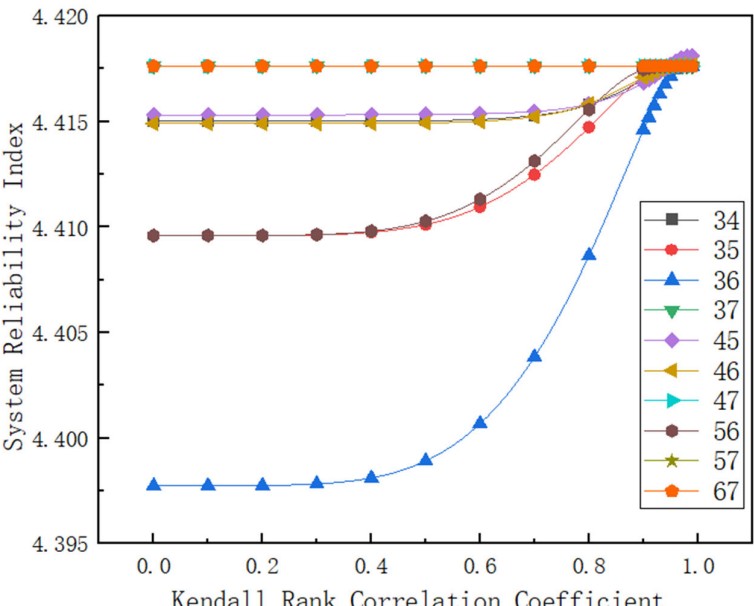

**Figure 3.** Effect of the change of the failure modes combination correlation coefficient on the reliability index of the system.

In order to more clearly describe the influence of Kendall rank correlation coefficient of each combination in the overlapping part of Figure 3 on the system reliability index, Figure 4 enlarges the influence of the change of the Kendall rank correlation coefficient of failure mode combination on the system reliability index when the change range of the Kendall rank correlation coefficient is 0.8~1.0.

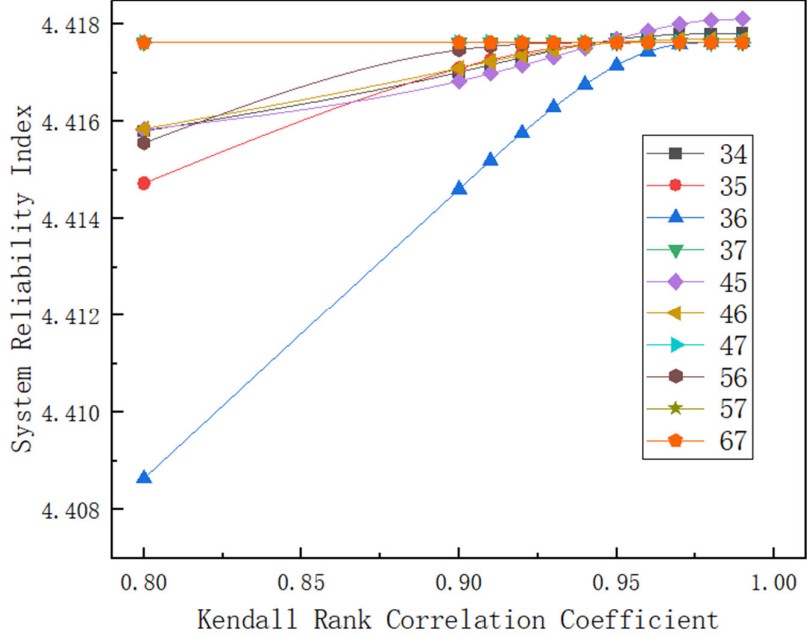

**Figure 4.** The influence of the change of the Kendall rank correlation coefficient on the reliability index of the system when the change range of the Kendall rank correlation coefficient is 0.8~1.0.

## 4. Conclusions

In this paper, based on an in-depth study of the existing relevant reference, a method of using Copula theory for structural system reliability analysis in the construction phase of bridge projects was proposed, and the study mainly achieved the following conclusions.

(1) The theory and method of calculating the reliability of the traditional system of bridge structures were summarized in the light of the existing research, and the shortcomings of the traditional theory in calculating the reliability of the system were analyzed. The idea of using Copula theory to establish the correlation between failure modes was proposed. (2) The basic contents of Copula theory were introduced in detail, and the Copula theory was extended to the field of reliability calculation of bridge structural systems. (3) The method proposed in this paper was used to calculate the examples in reference [34], and the formula of Copula theory was appropriately improved according to the calculation results. After comparing the results with the original reference, it was concluded that the improved Copula theory for calculating the reliability of the bridge structure system overcomes the problem of safety bias of the traditional Copula theory, and can be used with the traditional "interval estimation method", which is simple to calculate, easy to understand, is suitable for large and complex bridges, and has a certain guiding effect on engineering practice. (4) Finally, the sensitivity analysis was carried out from two aspects, and the results showed that the influence of the correlation between the failure modes on the failure probability of the structural system must not be ignored in engineering.

This paper has achieved milestones in the study of the reliability of bridge structure construction process system using Copula theory, but there are still areas that need to be considered and improved. Based on the research process and results of this paper, the general directions of the subsequent research are as follows. (1) In this paper, only the correlation between two failure modes was established, and the analysis was carried out from the perspective of two-dimensional Copula function, which is not closely enough connected with the actual engineering. The Copula function with a higher dimension should be used for calculation as much as possible so as to improve the connection with the actual project. (2) In this paper, only a coefficient was generalized into the improvement of the calculation theory to weaken the situation that the same failure mode appears many times in the calculation, and this calculation is not accurate enough. The weights of all failure modes and their combinations should be calculated and the weights brought into the calculation theory as coefficients so as to further improve the accuracy of the calculation results.

**Author Contributions:** Conceptualization, Q.L.; methodology, T.Z.; validation, T.Z.; formal analysis, Q.L.; resources, Q.L.; data curation, T.Z.; writing—original draft preparation, T.Z.; writing—review and editing, Q.L.; supervision, Q.L. All authors have read and agreed to the published version of the manuscript.

**Funding:** This research received no external funding.

**Institutional Review Board Statement:** This study did not require ethical approval.

**Informed Consent Statement:** This study did not involve humans.

**Data Availability Statement:** All the data used in this study can be found in the article.

**Conflicts of Interest:** The authors declare no conflict of interest.

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
