# Peer review of "Research on the Reliability of Bridge Structure Construction Process System Based on Copula Theory"

_applsci, doi:10.3390/app12168137_

Round 1
Reviewer 1 Report
- 1- The implications of AIC criterion and BIC criterion values need to be defined
- 2- Proper English review for the entire paper is needed. For example:
o Line 29, “steel-concrete combination continuous girder bridges” Should be “steel-concrete composite continuous girder bridges”.
o Line 29-31: “The bridge 29 construction phase, as one of the crucial steps in bridge construction, will directly deter-30 mine the quality and use of the bridge after it is completed.” This statement is unclear and needs to be reviewed and rephrased.
- 3- Equations and formulas need to be in a separate line with the proper numbering.
- 4- In-text citation needs to be revised (e.g. reference # 26 in Line 300)
Reviewer 2 Report
The manuscript “Research on the Reliability of Bridge Structure Construction Process System based on Copula Theory”, by Li & Zhang, presents an interesting procedure for the reliability assessment of bridge structural components during construction phases. Specifically, the authors apply the Copula theory to estimate the reliability index during the construction process. This proposed approach is also compared to the existing state of the art, i.e. the point estimation method and the interval estimation method.
Overall, the paper is of potential interest to readers of Applied Sciences. Nevertheless, some aspects, concerning both the paper’s content and its form, should be addressed before full acceptance. These are:
Main Issues:
1. Page 8: the acronym AIC (Akaike information criterion?) should be written in full the first time the acronym is introduced. Same for BIC (Bayesian information criterion ?). Moreover, the sentence at lines 268-269 seems odd; perhaps the intended meaning was "(1) AIC criterion and (2) BIC criterion"? In any case, more context should be given to the theory of both AIC and BIC.
2. Page 9: Even if, in general, is a good practice to refer to the existing literature (here, [26]) and avoid repetitions, here, the case study application should be fully reported in detail. That is because the referred document is an M.Sc. Thesis (thus, a non-peer-reviewed document), and also it does not seem to be publicly available on the internet. Furthermore, some more details about the 28 stages of construction can be (briefly) recalled as well, to make the discussion more self-contained.
3. The use of statistical tools to predict potential structural failures is of the foremost importance for bridge management in particular and for all kinds of civil structures and infrastructures in general. In this regard, however, the common practice of today generally includes continuous monitoring approaches. This relevant aspect seems to have been not mentioned in the paper introduction. In particular, some recent applications for civil structures have been recently proposed in the literature, e.g. in https://doi.org/10.3390/app11041716. the authors may consider adding this discussion to their paper.
4. The construction phases (quite minimally) described in Section 3 seem to refer to a prestressed RC bridge. However, as pointed out by the authors at the beginning of their Introduction, there are several typologies for railway and roadway bridges, including different building materials (e.g. steel) and structural designs (" large span steel pipe concrete arch bridges, large span suspension bridges and steel-concrete combination continuous girder bridges "). How would the proposed approach perform for these other bridge typologies?
5. From Table 7, the difference in joint failure probability for some cases is more than one order of magnitude (10^-17 vs 10^-6). The authors may add some more comments about this very large divergence.
6. The structure of the Conclusions, very schematic and methodical, is praiseworthy. However, they are still a bit too long. Shortening could be useful.
Editorial/Minor Issues:
1. Some inline equations, such as page 2 line 67, or page 5 lines 202 to 208, could be better highlighted as numbered equations.
2. Please double-check for grammar errors and typos, e.g. in the abstract: "afterward, an example is used" (without “then”); in the conclusion, “[..] suitable for large and complex bridge. And has a certain guiding [..]”, there are several mistakes: (1) the full stop mid-sentence is unnecessary; (2) bridges should be plural.
3. Table 1 (page 7): is it necessary to introduce the new variables \Zeta1 and \Zeta2 (greek letter) if these are only a rewriting of already existing variables (\Phi^-1(u) and \Phi^-1(v))?
In the same table, second row: the intended meaning of the <dot> function in ‘u <dot> v’ should be more clearly stated.
4. Table 4, for consistency, uses the same number of decimal digits everywhere in the same column (e.g. 0.0096 -> 0.00960, 0.9907 -> 0.99070.
5. Caption of Table 6: the meaning of the values highlighted in bold should be recalled.
6. Figure 1: a zoom box between xlim([4 8]) and ylim([4.0 5.0]) may help to better visualise the minor differences between the seven lines.
7. Figure 2: as above, a zoom box between xlim([0.8 1.0]) and ylim([4.415 4.420]) might be useful.
Reviewer 3 Report
A method of using the Copula theory for structural system reliability analysis in the construction phase of bridge projects is proposed according to the previously published papers. Based on the Copula function theory, the reliability of the construction process of bridges was analyzed. There are several comments as follows:
1- Abstract needs to be revised and be more straightforward toward the objective and results of the manuscript.
2- comprehensive national power should start with capital letters?
3- The second paragraph is too long. Please change it to two or three paragraphs.
4- Page 2 Line 63, reference 2 is not related to the PENT method. Please revise.
5- Please define each abbreviation in the first occurrence in the context. See Page 3 Line 129.
6- Please revise Lines 138-145 on page 3 as it is hard to follow for readers.
7- The title of 2-2-1 and 2-2-2 are the same.
8- Page 6 line 224, revise the sentence.
9- Please clarify the failure criterion for failure modes in Table 2.
10- Please explain that there might be some other uncertainties and also add how you or the other literature select uncertainties based on Table 3.
11- Please specify what is the limit for not considering any correlation between two variables in equation 17 based on Table 4.
12- Page 15 Lines 441 to 449, need to be rewritten.
13- Please clarify what is the main reason for using the proposed method to calculate the reliability index as there are several methods available in the literature.
14- Page 16 Line 481, the difference between the influence of correlation coefficient of 36 and 67 is about 0.5%, so it is not great as you mentioned in the manuscript.
15- What is the reason for these two sentences: "he combination of failure mode 3 with failure mode 5 and the combination of failure mode 5 with failure mode 6 have a greater influence on the reliability indexes of the bridge structure system " and "the combination of failure mode 4 with failure mode 5, failure mode 4 with failure mode 6, and failure mode 3 with failure mode 4 have less influence on the reliability index of the bridge structure system "? It should have a reason in the real world.
16- Using recent references are strongly suggested.
Round 2
Reviewer 2 Report
The authors have adequately faced the main observations raised by this reviewer, both regarding the conceptual aspects and the format of the paper. There are only a few remaining editorial issues, which should be fixed:
1. Table 5 has the same issue noticed in the first round of review for Table 4: the first 4 rows of the second column have 4 decimal digits, while the last and second-to-last ones have five (0.99989 and 0.99986). Please double-check carefully for all Tables.
2. The intended meaning of the last sentence of Section 3 (“This is consistent with the previous content.”) is not very clear.
3. In the newly-added Eq (1), the meaning of ‘P_f’ and ‘P_fl’ should be reported in the text.
Reviewer 3 Report
The manuscript is ready to be published.
Author Response
Thank you very much for your affirmation of our work.